# Targeting the Human Influenza a Virus: The Methods, Limitations, and Pitfalls of Virtual Screening for Drug-like Candidates Including Scaffold Hopping and Compound Profiling

**DOI:** 10.3390/v15051056

**Published:** 2023-04-26

**Authors:** Thomas Scior, Karina Cuanalo-Contreras, Angel A. Islas, Ygnacio Martinez-Laguna

**Affiliations:** 1Faculty of Chemical Sciences, Benemérita Universidad Autónoma de Puebla, Ciudad Universitaria, Colonia San Manuel, Puebla 72570, Mexico; 2Vicerrectoría de Investigación y Estudios de Posgrado, Benemérita Universidad Autónoma de Puebla, Puebla 72592, Mexico

**Keywords:** influenza, neuraminidase inhibitors, noncompetitive inhibition, virtual screening, ligand docking, screening pitfalls, screening problems

## Abstract

In this study, we describe the input data and processing steps to find antiviral lead compounds by a virtual screen. Two-dimensional and three-dimensional filters were designed based on the X-ray crystallographic structures of viral neuraminidase co-crystallized with substrate sialic acid, substrate-like DANA, and four inhibitors (oseltamivir, zanamivir, laninamivir, and peramivir). As a result, ligand–receptor interactions were modeled, and those necessary for binding were utilized as screen filters. Prospective virtual screening (VS) was carried out in a virtual chemical library of over half a million small organic substances. Orderly filtered moieties were investigated based on 2D- and 3D-predicted binding fingerprints disregarding the “rule-of-five” for drug likeness, and followed by docking and ADMET profiling. Two-dimensional and three-dimensional screening were supervised after enriching the dataset with known reference drugs and decoys. All 2D, 3D, and 4D procedures were calibrated before execution, and were then validated. Presently, two top-ranked substances underwent successful patent filing. In addition, the study demonstrates how to work around reported VS pitfalls in detail.

## 1. Introduction

Despite world-wide vaccination efforts and “anti-flu” public health prevention campaigns (general hygiene and patient confinement), the influenza disease has never been controlled due to antigenic drifts or occasional abrupt shifting by gene mutations in the viruses—in addition to viruses occasionally crossing host–range barriers, thereupon expanding their genetic pools. Hence, as a severe setback, vaccines must be developed in advance of a forthcoming flu season based on predictions on previously known strains. Moreover, vaccine production time is a bottleneck, so vaccines are not readily available during the initial spread of a pandemic. As such, alternative treatments such as novel mini-antibodies have been proposed [1]. Human influenza A vaccines constitute the cornerstone of flu prevention while drugs have been developed for monotherapeutic purposes in the early stages of infection or as a co-medication, such as orally inhaled zanamivir (Relenza™) or orally administered oseltamivir (Tamiflu™). The latter suffered from complicated multi-step and resourceful synthetic preparation challenging production and delivery on time. These setbacks beg for simpler chemical moieties with broad anti-flu activity (against H1N1 and H5N1), as well as the possibility of structural derivatization.

Traditional experimental drug development techniques are based on laborious optimization cycles until favorable results are met. In contrast, computational molecular simulations—namely virtual screening, scaffold hopping, and drug candidate profiling—can save time, materials, and human resources. In particular, compound collections allow the inclusion of huge amounts of chemical substances—either already extant “real world” substances or non-existing molecules from virtual libraries. In this context, novel scaffolds against new influenza virus targets have emerged [2].

Three types of viral proteins are located on the influenza virus’ surface: M2, HA, and NA. Primary sequence variants of the latter two define the subtypes, including human H1N1 or avian H5N1, etc. The human influenza A neuraminidase (protein name NA, gene name: na) is similar to the other viral surface glycoprotein hemagglutinin (HA), with complementary functions during the infection process. NA and HA protrude spike-like above the viral outer envelope, where four NA glycoproteins usually form homotetramers [3]. Mechanistic insight concerning virus-binding specificities to sialyloligosaccharides on human cells from clinical isolates was published as early as 1989 [4]. Of all nine existing NA subtypes, only influenza virus types A, B, and C circulate in the human body, specifically N1 from influenza A virus [5,6]. Originally H5N1 was an avian influenza subtype, but, after the 2009/10 “bird flu”, this virus has been suspected to be on the brink of zoonosis, risking a viral pandemic spread among humans [7]. Thus, the N1 protein is a prime and timely target for antiviral screening.

Human influenza viral neuraminidase extracts sialic acid (N-acetylneuraminic acid) from the sugar chains of glycoproteins on the human cell membrane surfaces. During initial infection and subsequent reproduction cycles in the patients’ mucosa cells, this viral sialidase has two essential functions: (i) it breaks down the mucin in the mucus layer, thereby facilitating viral entry into upper respiratory epithelial cells; (ii) it enables the dissemination—i.e., the “budding”—of newly reproduced virus particles from the host cells. During the last phase of budding, when the new viruses are still attached to host cell glycoproteins, galactose and sialic acid constitute the last two sugar monomers in the sialoglycan receptor on the human cell membrane surface. Both are linked by a glycosidic bond that is enzymatically cleaved by viral NA during budding when detaching offspring viruses from the human host cell. In contrast, viral HA binds to the terminal sialic acid residue of human sialoglycan receptors during the initial stage of infection to attach the incoming virus particle to the host cell surface. However, this HA-mediated virus–host binding can interfere during the late infection cycle when the multiplicated new outgoing offsprings are prevented from escaping the cell surface. Thus, the virus requires the enzymatic intervention of its neuraminidase to enable its separation from the host cell. The viral neuraminidase belongs to the glucosidase family (EC: 3.2.1.18), and has an aspartate and glutamate catalytic dyad in the active site that is responsible for cleaving the glycosidic bond between the hosts’ sialic acid and galactose. Precisely, Glu277 attacks the glycosidic bond, forming a cationic intermediate (a carbenium-oxonium moiety) in noncovalent concert with the adjacent Tyr406 [8,9].

The present study describes not only the findings in details, but also the methods that we used to find drug-like molecules combining computational screening, docking, and profiling. We also discuss some of the limitations of virtual screening and the pitfalls to be avoided during virtual screening. Our approach is exemplified by a description of our discovery of small Fmol and AAmol molecules that are predicted to inhibit the budding of influenza viral particles. The target biomolecule belongs to the human influenza A H1N1 virus from the 2009 pandemic outbreak, also known as “Mexican flu” (A/California/04/2009 (H1N1)), as well as the H5N1 subtype, also known as “bird flu” (A/Viet Nam/1203/2004 (H5N1). In the meantime, clinical observation has revealed that H5N1 infection is less common in humans. The motivation for searching for a combination of virus types is intended to identify conserved scaffolds common to all N1 proteins that are unlikely to possess intrinsic target preferences or exhibit mutational resistance, i.e., complications which lie beyond our scope and are addressed by others [10,11]. Our study is embedded in the extant literature on ongoing drug research to find new scaffolds for targeting neuraminidase N1 [11,12].

Prior to virtual screening (VS), a pharmacophore model was created upon assessment of the common binding pattern. To this end, we inspected the known binding modes of four commercial influenza virus N1 inhibitors (oseltamivir, zanamivir, peramivir, and laninamivir). Sialic acid was also included in the study as it constitutes the natural component of host cell membranes, which is recognized as substrate by the viral neuraminidase enzyme. Also under scrutiny was structurally related DANA (Appendix A). In addition, a complete ADMET profile was generated for the two proposed drug-like hits on theoretical ground.

## 2. Materials and Methods

### 2.1. Computer Programs

The following computer programs and web-based general tools were used in this study: Vega ZZ [13], Pubchem at https://pubchem.ncbi.nlm.nih.gov/ (first accessed on 24 April 2013) [14], Autodock v. 4.2 with AutoDock tools (ADTs) [15], Discovery Studio v. 4.0 [16], and OpenBabel [17]. In addition, two specialized tools were used for the VS and ADMET profiling of VS hits: Molecular Operating Environment [18] and ADMET predictor v. 7.1 [19] at https://www.simulations-plus.com (first accessed on 29 August 2014).

The Brookhaven Protein Data Bank at http://www.rcsb.org/pdb (first accessed on 5 September 2013) [20] was visited to download the target structure N1 [21]. Moreover, we retrieved crystallized structures of the liganded N2 complex with sialic acid (PDB entry: 2BAT) [22], oseltamivir (PDB entry: 3CL0) [23], zanamivir (PDB entry: 3TI5) [24], laninamivir (PDB entry: 3TI4) [24], peramivir (PDB entry: 4MWV) [25], and substrate-like DANA (PDB entry: 2HTR) [26] as reference ligands with known binding modes and activities.

The water-accessible surface of the N1 active site was calculated at 4.5 Å and potential binding patterns with characteristic chemical properties (features) within the site were identified, i.e., surface locations to form hydrogen bonds, salt bridges, polar regions, and hydrophobic pockets.

The study design embraced four main procedures: (1) pharmacophore modeling, (2) drug-like dataset screening, (3) affinity docking, and (4) pharmacokinetic profiling.

1.To determine the pharmacophore patterns, we studied binding mode specificities and structure–activity relationships (SAR) of hitherto known 3D structure complexes between influenza virus neuraminidase targets and sialic acid substrates or four reference antiviral drugs (oseltamivir, zanamivir, laninamivir, and peramivir) to generate 1D, 2D, and 3D fingerprints used as filters during virtual screening (VS).2.We also aimed to carry out VS on drug-like compounds by 1D, 2D, and 3D fingerprints. The method facilitates a fully automated (unsupervised) selection of drug-like candidates. To this end, 1-, 2-, or 3D filters have to be predefined or are built-in search tools [27,28,29,30]. The input data collection comprises a total of 660,961 small organic molecules (SOMs) [18]. It is composed of basic commercial structures for next-step experimental lead optimization and scaffold diversification upon identifying selected VS hits.3.The ligand affinities were then calculated to target for the selected VS hits by molecular docking. The self- or back-docking of reference inhibitors compares their predicted poses and affinities with observed (crystallographic) data and validates the computational study, besides docking new molecules to target. The molecular affinities were compared to the viral neuraminidase target by molecular docking each of the following ligands: natural substrates, sialic acid, reference drugs, and VS hits. Interaction energies and affinities were quantitated by means of the inhibition constant (K_i_), and the results of the computed affinities were compared with experimental K_i_ values of reference drugs from the literature.4.For ADMET profiling, smile codes were created for the VS hits and ADMET data were assessed using ADMET Predictor software.

### 2.2. Virtual Screening 

The search for new candidate drugs through VS can be carried out with in-house molecular databases or public virtual chemical libraries, both of which may store large amounts of molecules and biological metainformation [31]. Alternatively, researchers took a closer look at natural sources for agents against the flu [32]. Here, a commercial substance library with a total of 660,961 chemicals was screened [18]. To cope with the sheer number of data entries, a stepwise approach was established with four search levels by applying different data-type complexity and filtering conditions (Figure 1).

i.Based on molecular overall features, thousands of chemical substances were eliminated by of their size (molecular weight), lipophilicity (log P), and toxicity (toxic groups, SMILES patterns). Such screening methods are termed one-dimensional (1D) filters.ii.All molecules which passed the 1D filter were filtered through topological searches for 2D binding patterns.iii.Utilizing active conformations of known ligands at the binding site, a pharmacophore 3D filter was designed and a conformational database of the remaining substances was searched for spatial matches (hits) of atoms, groups, or properties (acidic, basic, polar, nonpolar, ionic, H-bond etc.).iv.Finally, the few 3D filter hits were screened by docking simulations, also sometimes called 4D filtering.

Molecular fingerprints are 2D or 3D binding patterns with numeric thresholds to determine whether dataset molecules pass or fail these 2D or 3D filters. We used the chemometric feature known as the Tanimoto coefficient to group or cluster similar molecules in case they exceeded the established similarity threshold (cf. user guide [18]).

To verify if VS can successfully discriminate between target binders and non-binders, we applied retrospective virtual screening: we enriched our dataset with a small number of compounds with known antiviral activities (positive controls) or non-activities (negative controls, also known as decoys). A total of ten positive controls were included: 2-deoxy-2,3-didehydro-N-acetylneuraminic acid (DANA), oseltamivir, peramivir, zanamivir, laninamivir, sialic acid, BANA113, BANA106, BCX1898, and G20. The first six were extracted from the aforementioned PDB files whereas the remaining four were taken from a published source [33]. In addition, 23 negative controls were added from PubChem [14]. First, the following numerical descriptors were calculated for the positive controls: molecular weight; hydrogen bond acceptors (HBAs); hydrogen bond donors (HBDs); acidic, basic, and hydrophobic atoms; and rotatable bonds (RB). Next, we searched PubChem for compounds with descriptor values similar to the positive controls but without reported activities against the neuraminidases of the influenza A virus.

The preparation for retrospective and prospective 2D virtual screening required four steps: (i) the removal of binding-irrelevant molecular components; (ii) protonation/deprotonation at physiological pH; (iii) the calculation of Gasteiger partial charges; and (iv) the calculation of TGD-type fingerprints.

The preparation procedures used for the virtual compound library—the Molecular Operating Environment (MOE) database for retrospective and prospective 3D virtual screening—differed only in the last step to calculate ligand conformations, whereby a stochastic algorithm was used under the MOE’s default force field MMFF94x [18]. Its applicability range not only covers drug-like molecules but also proteins. Three parameters were set as follows: (i) residual strain energy limits at the local minimum: 10 kcal/mol; (ii) the maximum number of conformations per molecule: 1000; and (iii) the root mean square distance (RMSD) threshold to remove duplicates: 0.15.

### 2.3. Molecular Docking

The target biomolecule was prepared for docking by removing undesired moieties from the crystal input structure using SPDBV. The protein’s atom partial charges on all amino acid atoms were computed using the Gasteiger approach of ADT.

All ligand models for docking were preprocessed under Vega ZZ in order to assign correct bond orders, hybridizations, hydrogen atoms, atom types (Tripos), and partial charges (Gasteiger), and the protonation/deprotonation state at pH 7.

For flexible ligand docking, default program settings were used under the Lamarckian genetic algorithm, except for the number of runs (256), elitism (3), and the highest precision level (25,000,000). The numerous peptide bonds were held rigid in their natural trans configuration, i.e., they were not allowed to rotate.

### 2.4. ADMET Profiling 

After screening and docking, pharmacokinetic characterization was carried out for the final candidates. Their experimentally confirmed antiviral activities are reported elsewhere [34].

Their 3D structures were converted into the SMILE code or mol file format under Vega ZZ, which were then used as input data for the ADMET predictor™ tool. The user manual describes the “expanded applicability domain” after including an in-house dataset from Bayer™. The improvements in the tool became evident “in prospective predictivity of S+ pKa certainly reflect [the tool´s] expanded applicability domain” [“…” cited from the ADMET predictor manual, 23 July 2014, version 7.1].

## 3. Results

### 3.1. Binding Pattern and Pharmacophore Modeling

The selected biomolecular target was the three-dimensional structures of the influenza A virus neuraminidase, corresponding to the viral strain A/Vietnam/1203/2004 (H5N1). As reference binders for target screening and docking, we retrieved the published crystal structures of four N1 inhibitor drugs in the complex with the target protein: zanamivir, oseltamivir, peramivir, and laninamivir (Table 1). In addition, the crystal complex of the N1 target with natural substrate sialic acid was also retrieved from the Protein Data Bank (PDB). Specifically, we inspected the following related PDB entries: 3CL2, 2HU0, 3B7E, 3CKZ, 2HTU, 2HTW, 2HTR, and 3NSS.

The fundamental tenet of (quantitative) structure–activity relationships declares that similar chemical structures reflect similar biological activities, although so-called activity cliffs create exceptions to the rule [35,36]. Therefore, we analyzed the binding modes at atomic scale of all five reference ligands from Table 1 (Figure 2).

All reference inhibitor modes of binding are fairly similar to the natural substrate sialic acid (Table 2). In the Appendix A, we document all details on ligand–receptor interaction patterns (Appendix A), which are in line with literature describing essential residue active conformations for ligand recognition [37]. To be more precise, ionic (and polar) head group atoms of Glu276, Glu277, Arg292, Asn294, and Ser246 mark the location of a large cavity. A smaller hydrophobic pocket is surrounded by Arg224 and Ala248. A larger pocket encompasses hydrophilic and hydrophobic residues Glu119, Asp151, Arg152, and Glu227. A fourth, negatively charged pocket is composed of residues Arg118, Arg292, and Arg371 (Appendix A).

The X-ray structure of the neuraminidase subtype N2 with co-crystallized sialic acid indicates that the substrate binds the enzyme in a considerably deformed conformation due to strong ionic salt- and hydrogen-bonding energies exercised through the substrate’s anionic carboxylate group. The latter is in contact with three cationic side chains of Arg118, Arg282, and Arg371. The N-acetyl group, attached to ring atom C5, maintains polar and nonpolar contact with Arg152, Trp178, and Ile 222. These interactions help anchor the substrate to the active site. The C4-OH group of sialic acid hydrogen bonds with the negatively charged Glu119, Glu227, and Asp151. In docking simulations of the natural substrate sialic acid, we analyzed the pivotal interactions in the N1 target active site. Remarkably, there is a concert of strong salt bridges and elaborated networks of polar hydrogen bonds between the ligand’s anionic carboxylate group and three cationic arginine residues (Arg118, Arg282, and Arg371). In addition, a fourth arginine Arg152 interacts with the substrate’s acetamide group. The stabilizing noncovalent hydrogen bond is formed between anionic glutamate Glu276 and two hydroxyl groups on Carbon atomsC8 and C9 of the ligand´s triol side chain. Of note, Glu276 is conserved in N1 and N2.

It is evident that all five sialic acid analogues share an aliphatic ring, a carboxylic group, an acetamide, and hydroxyl groups or other oxygenated functions for hydrogen bond networking. These findings were merged with other findings into the 2D and 3D fingerprint (pharmacophore) models. Appendix A exemplifies the graphical analysis for oseltamivir. Based on the calibration results, pharmacophore model 24 was chosen to perform the prospective 3D virtual screening. Stereochemical drawings of the reference ligands (Appendix A) were aligned (superposition) to generate 3D pharmacophore models (Figure 3; shown in detail in Appendix A). This allowed us to discriminate all 10 active molecules from the 23 inactive decoys. Moreover, model 24 obtained the best metrics score during evaluation.

### 3.2. D Filtering (2D Fingerprint Design) 

Several fingerprint models were constructed from the two-dimensional features of the following ligands: sialic acid (2BAT), oseltamivir (3CL0), peramivir (2HTU), zanamivir (2B7E), and DANA (2HTR). The fingerprint filter is two-dimensional, so the conformations of the molecules and the RMSD between them were not included. The molecules were processed, as indicated in the Methods section. Subsequently, the 2D typed graph distance (TGD) fingerprints of the chosen molecules were calculated.

During retrospective virtual screening, all fingerprint models were constructed and underwent repeated rounds until an optimized outcome from the control test set was achieved. Finally, one best-scoring model was selected, i.e., that with the highest hit rate to discriminate between the active and inactive compounds.

### 3.3. D Filtering (3D Fingerprint Design)

To determine the essential set of 3D structural features to include in the pharmacophore models, the reference ligands underwent superposition operation, the RMSD values were calculated, and the overlapping features were deemed pivotal. These “frequently conserved” ligand substructures were merged to formulate the pharmacophoric models such that the several initial models were extracted from the co-crystallized ligands (with PDB codes): sialic acid (2BAT), oseltamivir (3CL0, 2HU0), peramivir (2HTU), zanamivir (2B7E), and sialic-acid-related DANA (2HTR). Since 3D substructures had been extracted from (3D) crystal data, all binding-relevant features could be located in 3D spaces (i.e., hydrophobic, anionic, and cationic chemical substructures) or hydrogen bond donor or acceptor (HBD or HBA, respectively) groups. Again, pharmacophore models were repeatedly evaluated against the enriched dataset by retrospective virtual screening. The model with the best activity vs. nonactivity discrimination capacity (model number 24) was finally selected.

### 3.4. Prospective 2D VS 

The final 2D fingerprint model was applied to screen the 2D data-type version of the MOE database [18]. This approach reduced the original ~10^5^ chemical entities of the MOE database to a smaller set by a 2D filter, which runs many thousand-fold faster than 3D filtering (Figure 1). 

### 3.5. Prospective 3D VS 

Pharmacophore model 24 was applied as a 3D filter to screen the 2D VS hits in order to obtain a final set of ~10^1^ candidates after clustering. To this end, the conformation of the lowest RMSD value with respect to its underlying pharmacophore model was computed for each 3D hit. Specifically, the TAT-type fingerprint of each 3D hit was calculated. The hits were lumped together using the Tanimoto coefficient, and the molecules that exceeded an arbitrary 85% similarity threshold to other candidates (with an identity match set to 100%) were not included for further study.

Fully automated multi-step VS led to a considerable data size reduction (Figure 1). After screening, two molecules of the 30 remaining were of limited presence in academic literature, and thus were worthy of further scrutiny. Hence, the molecular docking of 30 candidates against the viral target led to two high-scoring candidates (Fmol and AAmol). We kept their nicknames: AA stands for two amino acids and F stands for the other (dimethyl-) phenyl group (written in Spanish as “Fenil”) (Figure 3). In collaboration with a biomedical research center (CIBIOR, Metepec, PU, Mexico), they were experimentally validated as active inhibitors and published together with other inhibitors from studies on drug repurposing or structure–activity relationships [34].

AAmol, with the chemical name (N-acetyl-phenylalanyl)-methionine, possesses two amide bonds. It is a dipeptide derivative (Phe and Met), has a total charge of −1 as it takes a deprotonated carboxylate form in water at pH 7, and interacts with arginine residues at the binding site. Its molecular mass is 337 Daltons; with an estimated log P = 1.2; a polar surface area of 183 Å^2^; a non-PSA of 441.9 Å^2^; a HBA = 4; HBD = 2; highly flexible conformations (12 rotatable bonds); and a commercial vendor: Chembridge # 6429718. A recent review outlined the fundamental principles concerning peptide inhibitors against the flu [38]. One 24-residue-long oligopeptide binds NA with a micromolar inhibition constant (Ki = 0.29 mM), and a much smaller octapeptide was designed as a strong nanomolar binder at the active site where oseltamivir appears, showing one-digit micromolar inhibition activity in the cell tests. In this context, AAmol, with only two amide bonds, is an even smaller H1N1 NA inhibitor than all the other reported peptides (4 to 24 residues long) [38].

Fmol, or 3-[(2,5-dimethyl phenyl) carbamoyl]-2-(piperazin-1-yl) propanoic acid, has two rings and also contains a monoanionic carboxylate group to interact with the cationic active site arginine residues. The piperazine ring possesses two potential protonation sites and exists as a di-cationic species in strongly acidifying media, but the monocationic form dominates under cellular conditions. With a mono-anionic and cationic center, it is a sort of zwitterion at pH 7, with a molecular weight = 306 Daltons; a log P = −3.4; a polar surface area = 142 Å^2^; a non-PSA of 427.9 Å^2^; a PSA-to-non-PSA ratio = 1:3; a HBA = 4; a HBD = 3; flexible conformations (seven rotatable bonds); and a commercial vendor: Life Chemicals F1278-0516.

### 3.6. Virtual Library Performance under Fingerprint Model Number 24

Starting with 660,961 entries in the MOE database, the 2D screening yielded 4853 hits (0.7% of the drug library), whose subsequent conformational calculations were then further virtually screened through the 3D fingerprint filter (for details, cf. Appendix A concerning dataset test characteristics, control structures and decoys, dataset enrichment metrics, further metrics on positional fit by RMSD, as well as pharmacophore modeling and thus the associated performance).

In order to score and evaluate the post-3D-filter hits they were docked against the influenza A virus N1 neuraminidase receptor. For this post-screening ranking by docking scores only a numeric evaluation of the hits was requested. Specifically, an inspection of conformational space and steric requirements was not intended for that stage. The study did not focus on the search for favorable positions or conformations (since the 3D filter yields very specific steric, electronic, and spatial conformations), but rather on the ranking of selected hits.

Successful self- or back-docking tests gave reason to assume that the blind docking of unknown active conformations of the hit ligands will be faithful and biochemically meaningful (Appendix A). Validation took place with reference ligands extracted from liganded crystal structures (with the following PDB codes): sialic acid (2BAT), oseltamivir (3CL0), oseltamivir (2HUO), peramivir (2HTU), zanamivir (2B7E), and substrate analog DANA (2HTR). The hits were blind-docked and ranked. The computed inhibition constant of oseltamivir (K_i_ = 0.007 µM) lies within the experimental range reported with PDB entry 3CL0 (0.0001 to 0.008 µM). We add another proof of concept that uses K_i_ values for our docking approach (for details on docking precision, see the Discussion section below). The docked ligand successfully reproduces specific contacts from the experimental structure—between the acetamido function and conserved arginine Arg152—despite the changes in docked overall positions which could be expected from scaffold hopping in our analyses (Appendix A) [39].

### 3.7. Ligand–Target Docking

The 3D models were curated in the Vega ZZ program [13], and their ionization states were estimated. A total charge of −1 can be attributed to both AAmol and Fmol (each with one carboxylate group), while a zwitter form (+1/−1) was ascribed to Fmol, i.e., an additional tertiary ammonium next to the carboxylate function. At pH 7, the piperazinyl ring is a mixture of monoaninic > neutral >> dianionic species following the rule of thumb for the dissociation of (diluted) weak acids or bases: pKa-pH yields the % concentration for neutral bases (or deprotonated corresponding acidic forms, respectively). At that stage of work, partial charges were loaded by the Gasteiger approach under Vega ZZ and the Tripos force field atom types assigned prior to saving in the mol2 file format.

All five reference models could be successfully docked back into their crystal poses (Table 1 and Table 2). For sialic acid (PDB entry: 2BAT), no experimental binding affinities have been published, but experimental affinity data of the four others have been included in their PDB entries. The experimental binding constants ranged from 0.1 to 817 nM for oseltamivir (PDB entry: 3CL0) and from 0.5 to 12 nM for zanamivir (PDB entry: 3TI5). Their computed best-scoring values from our docking studies lie in the same one-to-three-digit nanomolar range (7.5 nM to 363.7 nM). Reported experimental IC_50_ values of peramivir (PDB entry: 4 MWV) and laninamivir (PDB entry: 3TI4) were 0.4 nM and 0.947 nM, respectively, which can be compared to their computed nanomolar inhibitory constants of 109.9 nM and 743.1 nM For the two hits under scrutiny, AD4 found the following affinities: (i) AAmol with a lower one-to-two-digit micromolar range; and (ii) Fmol with an upper nanomolar range, with the lowest K_i_ of 0.1 mM and an average value of 0.8 mM for the most populated, best-scoring (first) cluster. For more details, see the Discussion section below. The final poses of the blind docking of AAmol and Fmol were compared to the reference complexes (Figure 4 and Figure 5).

Figure 4 illustrates the equivalent role of three structural features which are shared by AAmol and oseltamivir. They effectively occupy the same cavity locations: (i) acetamido; (ii) alkyl-ether (-CH_2_-O-CH_2_-) and alkyl thioeter (-CH_2_-S-CH_2_-); and (iii) anionic carboxylate groups (-COO^(−)^). In the figure, both acetamide side chains (i) lie to the rear (topmost). The methyl group on AAmol (beige) is not visible, occluded by the acetamido group of oseltamivir which lies in the front. In the middle, both O/S-ether bridges (ii) are fully visible. The methyl-thioether group belongs to AAmol in the front (foremost, sulfur atom in yellow). Intriguingly, the three atoms of both carboxyl groups (iii) are in almost perfect superposition, presumably reflecting their role in ligand recognition as they help bind to the same cationic residues (Table 2), though only one oxygen atom from each of the COO^(−)^ groups can be seen (two overlapping rightmost red O atoms). CH-rich (aliphatic) substructures on both ligands squeeze to the left into the hydrophobic area. The extended lipophilic pocket remains unoccupied by both binders (cf. Val149, Ile 427, and Pro431 in Table 2). Figure 5 complements Figure 4 in additionally presenting Fmol and a top-down view on the entire carboxyl group. Their three atoms (-COO) appear perfectly aligned (Table 2). In the mid-section, the alkyl ether (oseltamivir) and thioether (AAmol) are fully visible as well. To the cavity’s side wall, N- and O-free alkyl parts on all ligands meet with more neutral residues (white surface), while the deeper, more hydrophilic rear (bluish surface) provides for more affinity for N- and O-rich substructures. The cavity entry coincides with the viewer’s perspective. Fmol’s dimethylphenyl ring extends between two hydrophobic areas in the forefront. In this way, it partly occupies the extended lipophilic pocket with Val149, Ile 427, and Pro431 (Table 2).

The interacting amino acids at the binding site of N1 were inspected in the 3D models with the docked poses and displayed in a 2D scheme (Figure 6 and Figure 7). In particular, AAmol’s docking shows that its acetamido group—which is part of the pharmacophore model—is correctly recognized by the Arg152 of the N1 protein target (Figure 6) [39]. Intriguingly, in Fmol, this amide bond is stabilized by a strong pi-electronic resonance effect since its nitrogen atom also belongs to the aromatic system of the dimethyl phenyl aniline substructure. Considering the latter as a sidechain and the remaining molecule part as the scaffold, the amide group orientation appears reversed to better fit into the cavity (Figure 5 and Figure 7).

### 3.8. ADMET Profiling

This theoretical pharmacokinetic profiling step was carried out as a post-screening evaluation of hits for early candidate attrition. In the context of long and costly drug research and development projects (R&D), in silico profiling studies are routinely carried out (Table 3).

## 4. Discussion

### 4.1. Implications and Limitations for Drug Screening 

Computational molecular simulation methods have ushered a new area of drug R&D in academia and industry. In silico approaches are popular, but currently generalized and in need of improvement. They are not intended to replace in vitro, ex vivo, or in vivo studies, but, rather, they should be understood as complementary tools within the drug discovery cycle. Pharmacophore modeling—specifically, molecular dataset screening for hits and blind docking of hitherto unknown target binders—is one such in silico approach which was validated herein by elaborated protocols, namely (i) analyzing binding patterns of published crystal complexes, (ii) fine-tuning of 2D and 3D filters and dataset enrichment with active controls and decoys, and (iii) back docking trials of crystalized ligand-target complexes. Bioassay results have been published to confirm the micromolar antiviral activities of our final candidates.

In general terms, VS saves time, material, human, and financial resources. However, this cutting-edge approach requires pre-existing information on drugs, compound libraries, and biomolecular receptors on an atomic scale. In short, as with any screen, the results are only as comprehensive as its inputs.

Although the filters used during VS had been selected after testing in the presence of known active and inactive control structures in order to perform a more selective screen, not finding hits with nanomolar target affinity does not necessarily jeopardize success, as high potency ought not to be expected in VS campaigns in general. Pragmatically, the ultimate goal of VS is to find hits or lead compounds for further research and validation by means of distinguishing between non-binders and binders—not necessarily between strong and weak binders (cf. pitfall (i) below). 

To the best of our abilities, this VS study concept is designed to avoid known pitfalls [40]. In particular, we worked around the following setbacks:

Pitfall (i): the success criteria of VS are insufficiently defined; solution (i): focusing on new scaffolds rather than high target affinities, which would be improved in a subsequent step of scaffold derivatization (drug profiling with design, synthesis, and testing).

Pitfall (ii): variable water-mediated binding interactions; solution (ii): in analogy with the situation illustrated in Appendix A Panel B with reference ligand zanamivir in interaction with a water moiety, we present the water-mediated AAmol interaction in Figure 6 with the graphical display of AAmol as a N1 binder.

Pitfall (iii): the rigorous and prospective validation of VS protocols; solution (iii): challenging our pharmacophore models with positive and negative control molecules in a virtual test library (Appendix A).

Pitfall (iv): overcautious approaches to ‘drug-likeness’; solution (iv): disregarding the “rule of five”, which is a retro-perspective result of averaged values on a drug sample (statistics), though it is not a prospective rule which must be followed (i.e., selection bias). As a direct result, Fmol is an unprecedent case of an aromatic-ring-bearing compound that is not seen on hitherto known N1 inhibitors. Moreover, the acetamido side chain still conserved on AAmol was incorporated into a larger substructure and its orientation was inverted to enhance cavity fitting (Figure 3).

Pitfall (v): one-at-a-time approach; solution (v): limited binding pattern complexity (2D or 3D filter definition) upon the application of a single active compound for pharmacophore generation.

Pitfall (vi): meaningless binding pattern selection for pharmacophore design; solution (vi): testing the filter capacity to discriminate between active and inactive control molecules. The former should appear in the hit list VS, while the latter should not, as also shown in solution (iii).

Pitfall (vii): the data size and structural variety do not reflect an appropriate variety of chemical spaces; solution (vii): the VS aimed to find new under-substituted scaffolds. Thus, possible hits with structural variations were searched in a chemical landscape of over half a million simple small organic compounds, as also shown in solution (i).

### 4.2. Implications and Limitations for Ligand–Target Docking 

Upon agonist or substrate binding, certain receptor types—such as nuclear steroid hormones, glucocorticoid receptors, or Cyp P450 enzymes—unfold side chains, backbones, or even domains in a phenomena called an induced fit [41]. In response to AD4’s treatment of flexible ligands and rigid body receptors, authors have proposed to act of docking apo-forms and various liganded complexes to account for spatial differences [42,43,44]. Others have used molecular dynamics to adopt new conformations by educated guessing [45,46]. None of the cited solutions were followed here for a two-fold reasoning: (i) the hits are all smaller in size than the six reference ligands; and (ii) the binding patterns of hits are merely a subset of those observed in the corresponding crystal complexes (Table 1). As such, no conformational rearrangements were needed.

AD4 computes free binding energies (ΔG) as a crude approximation from linear scaling on rotatable bonds of the entropy term (ΔS). The tool converts these thermodynamic estimates into inhibition constants (K_i_) by applying the thermodynamics formula K_i_ = e^[ΔG/(R*T) ]. Moreover, K_i_ values are not always close to IC_50_ values, which depend on the substrate concentration in the assay. The estimated K_i_ value for the endogenous sialic acid substrate lies in a two-digit micromolar range. Referring to the reference compounds, their micromolar K_i_ values all lie in one-digit ranges of the best-scoring clusters.

For AAmol, the experimentally determined inhibitor constant was published with a K_i_ value of 15 mM [34]. The best value in our docking simulation for AAmol was K_i_ = 0.001 mM, with an average value for K_i_ of 0.12 mM, which is approximately a 100-fold overestimation of potency. With regard to calculation precision vs. experimental data, the docking program tutorial states that the numeric results are merely crude estimates in a wider 100-fold imprecision range (corroborated by private messages from Autodock scientist Prof. Dr. Stefano Forli, Dept. of Comp Biol, Scripps Research Institute, La Jolla, CA, USA)—a fact which is overlooked all too often by published AD4 studies which claim that their numeric result lies in excellent keeping with assay data as a misleading “proof of concept”. Our proof-of-concept stems from the fact that the acetamido group on AAmol is recognized, as reported by the crystal complexes.

For Fmol, only a IC_50_ value was measured against N2, not N1. Given the experimental settings, the IC_50_ data of Fmol are not directly comparable to the K_i_ values. Of note, IC_50_ does not reflect directly binding affinities, but could be set on an equal footing by the Cheng–Prusoff equation, which is, unluckily, in need of additional experimentally determined parameters.

### 4.3. Implications and Limitations for ADMET Modeling

The predicted numerical results from ADMET profiling are crude estimates. The program’s artificial neural network architecture has come under criticism due to variable degrees of reliability. The latter depends on the applicability range of each ADMET parameter, which, in turn, is given by the training set for calibration, i.e., adjusting the outcome to some experimental (measured) endpoint. To complicate reliability, even closely related structures—e.g., warfarin and phenprocoumon—fall short of expectations (an unpublished pitfall during our phenprocoumon research [47]). One explanation for ill-behaving cases (pitfalls) is that overfitting tends to increment the prediction power, and in the same time narrows the applicability range to only those entries of the underlying training set used during data fitting. As a direct consequence, a trade-off must be arranged for an “in-between solution. Stray outliers lie in the nature of this method. At the essence of the pros and cons discussion here, it is safe to presume that VS results do not foresee pharmacokinetic issues with absorption, distribution, metabolism, excretion, or toxicity.

## 5. Conclusions

More than half a million small organic compounds from a commercial dataset were virtually filtered in four stages. One-, two-, and three-dimensional pattern filters were utilized and verified by enriching the dataset with molecules of known activity or inactivity, until the screen could distinguish between the active and inactive controls. A final (4D) filter was imposed with a ligand–enzyme docking for the best-performing hits from the virtual screen. Two such top-ranked substances underwent successful patent filing (details on the experimental validation having been published elsewhere). In line with the present study design and choice of input data, novel lead compounds were used with simple under-substituted scaffolds of micromolar target affinities, representing a promising future of ongoing R&D cycles of synthetic derivation and biochemical testing to develop stronger and more specific target binders.

## 6. Patents

The two concluded national patents have the following identification numbers: (i) for AAmol: MX/E/2017/039727, IMPI no. 352708 (2018); (ii) for Fmol: MX/E/2017/034353, IMPI no. 352709 (2018).

## Figures and Tables

**Figure 1 viruses-15-01056-f001:**
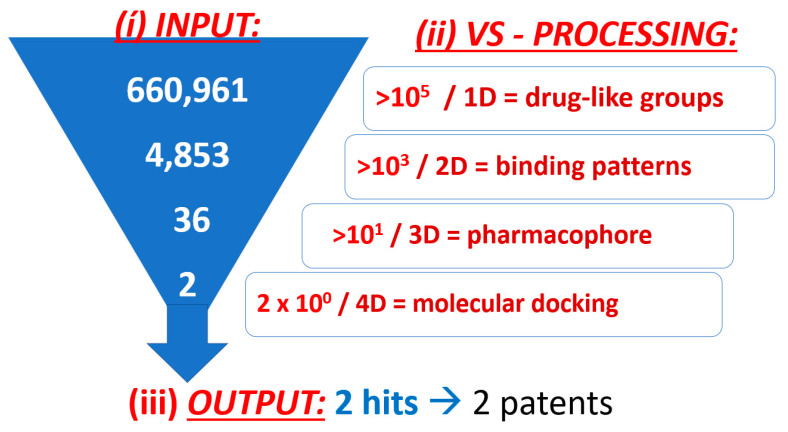
Schematic workflow of the virtual screen. This strategy filters over half a million substances of variable sizes and compositions. Positive and negative controls were manually added to the virtual library for screening. Due to funding limitations, only 2 out of 30 candidates could be experimentally examined for antiviral potency (0.005%, *n* = 660,961 initial entries). Both hits received patents as promising therapeutic leads. They constitute under-substituted scaffolds and are labeled AAmol and Fmol for short 10^3^.

**Figure 2 viruses-15-01056-f002:**
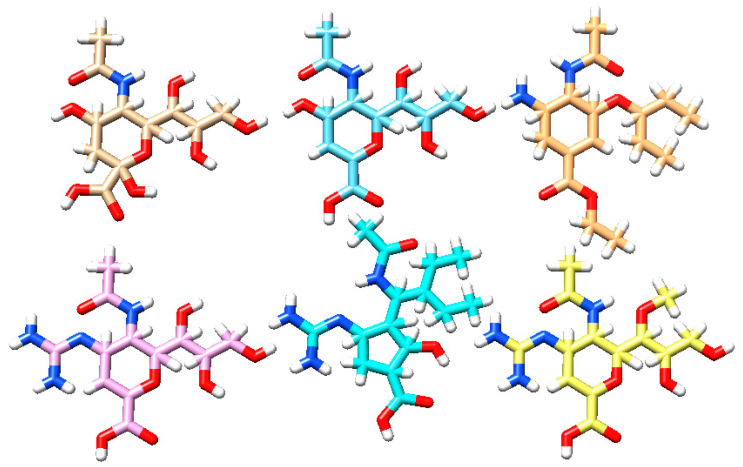
The endogenous substrate and five competitive reference ligands. The coloration and 2D orientation allows structural comparison by eye. Top row, left to right: sialic acid, DANA, and oseltamivir; bottom row, left to right: zanamivir, peramivir, and laninamivir. Color coding: white—H atoms, red—O atoms, blue—N atoms, and other colors—C atoms.

**Figure 3 viruses-15-01056-f003:**
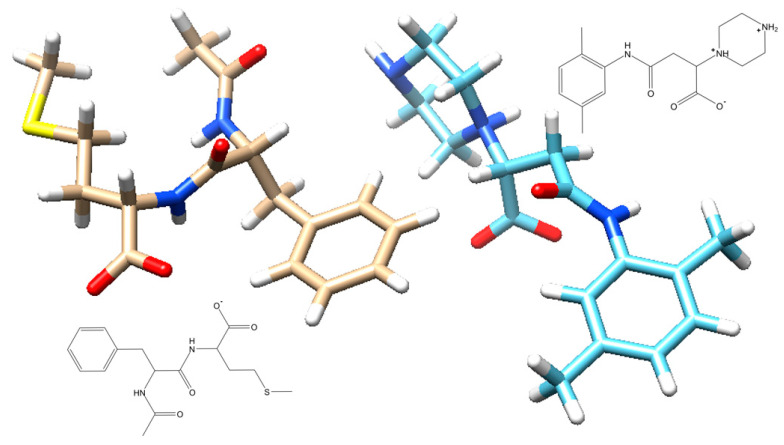
Three-dimensional stick models of the final hits with antiviral activity. AAmol (left) and Fmol (right) were selected among tens of final candidates from virtual screening. Fmol has two protonation siteson the tertiary and secondary amine groups on the piperazine ring. At physiological pH one is protonated while the carboxyl group is deprotonated. Two inlays show the structural 2D drawings of AAmol (below) and Fmol (above). The color coding for H, O, N, and S atoms: white, red, blue, or yellow, respectively. Carbon atoms are beige (AAmol) and light blue (Fmol).

**Figure 4 viruses-15-01056-f004:**
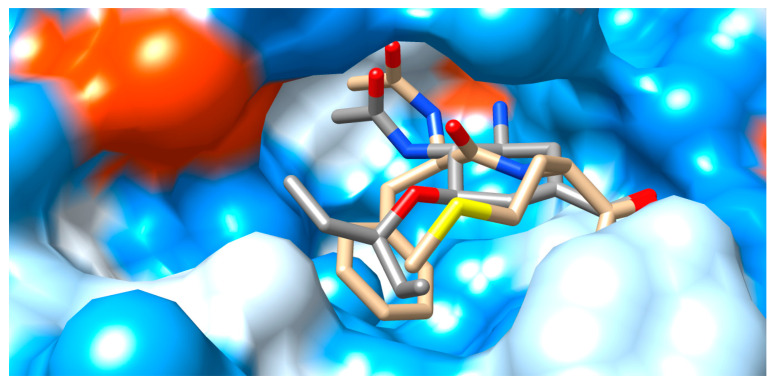
Structural binding model of AAmol (beige), with co-crystalized oseltamivir (grey) for reference. The cavity entry lies to the right hand coming from the foreground. Color coding for the stick models: red, blue, and yellow for O, N, and S atoms, respectively; beige or grey for carbon atoms of AAmol or oseltamivir, respectively; hydrogen atoms are omitted. The water-accessible protein surface is also colored: light blue for hydrophilic (polar) residues, orange for hydrophobic (nonpolar), and white for intermediate polarity.

**Figure 5 viruses-15-01056-f005:**
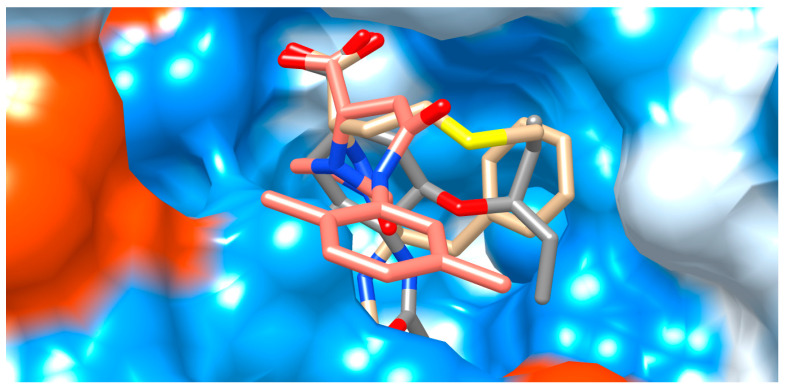
Structural models of binding modes for AAmol (beige), Fmol (pink), and co-crystalized oseltamivir (grey) for reference. Top-down view into the binding cavity. The viewing angle from Figure 4 is tilted 90° for orthogonal viewing. Coloration is shown in Figure 4.

**Figure 6 viruses-15-01056-f006:**
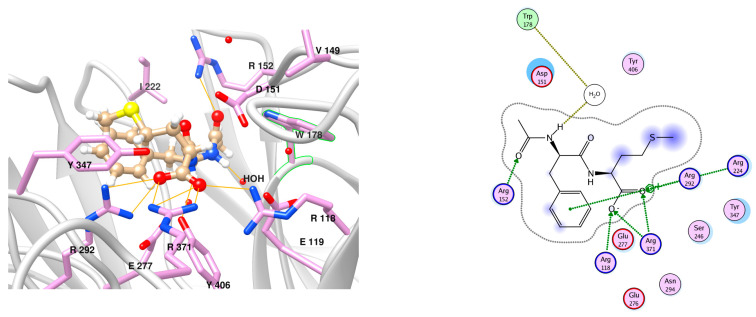
Schematic representation (**left**) and graphical display (**right**) of N1 binding site with docked AAmol (beige ball-and-stick model). The cavity entry lies to the foreground left. Color code for the stick models: pink, red, and blue indicates C, O, and N atoms, respectively; grey ribbons represent the protein backbone; residue hydrogen atoms are omitted; additional colors are shown in Figure 4.

**Figure 7 viruses-15-01056-f007:**
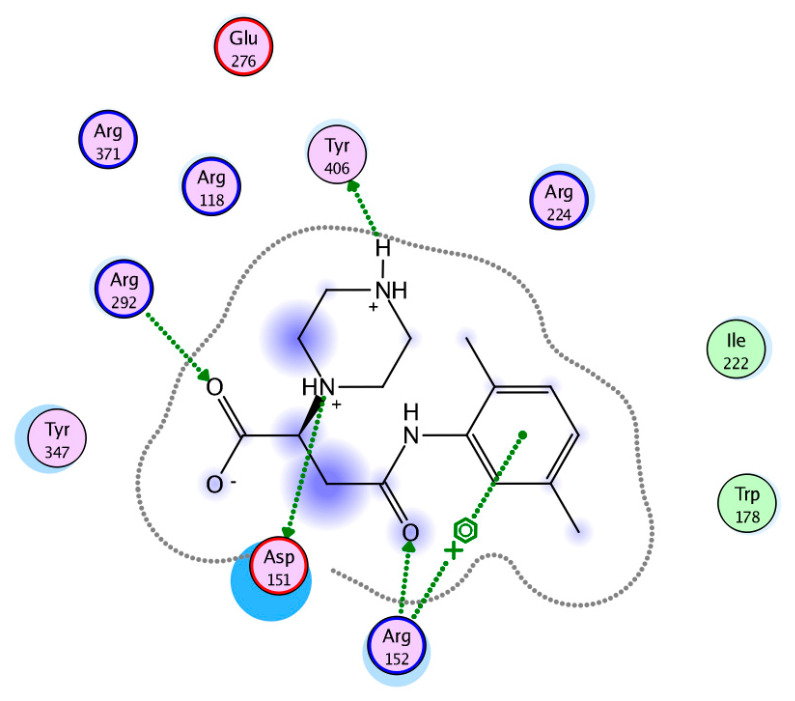
Graphical display of the N1 binding site with Fmol (pink). The cavity entry lies to the foreground left. Ile222 and Trp178 flank the hydrophobic dimethylphenyl side chain of the ligand to the right in the left-hand panel or to the left in the right-hand panel, respectively. Color coding is shown in Figure 4 and Figure 6.

**Table 1 viruses-15-01056-t001:** Input data from the human influenza A virus for 3D model generation from experimentally observed crystal structures of the reference target protein and ligands. Asterisk (*): the target primary sequence corresponds to a single-point mutant Iso223Val (I223V) or (**) His274Tyr (H274Y). As an off-scope effect, these changing amino residues also give knowledge about published resistance mechanisms [21].

Extracted 3D-Structure	PDB Code	Types of Molecules and Viruses	Resolution (Å); Year	Ref.
Neuraminidase (target protein)	5NZ4	OS—liganded neuraminidase N1; unidentified strain; (*)	2.2; 2018	[21]
Sialic Acid (SA)	2BAT	SA—N2 complex; influenza A virus; A/Tokyo/3/1967 (H2N2)	2; 1992	[22]
Oseltamivir (OS)	3CL0	OS—N1 complex; influenza A virus; influenza A virus; A/Viet Nam/1203/2004 (H5N1) (**)	2.2; 2008	[23]
Zanamivir (ZA)	3TI5	ZA—N1 complex; influenza A virus; A/California/04/2009 (H1N1)	1.9; 2011	[24]
Peramivir (PE)	4MWV	PE—N9 complex; influenza H7N9 virus; human-infecting variant from avian origin	2.0; 2013	[25]
Laninamivir (LA)	3TI4	LA—N1 complex; influenza A virus (A/California/04/2009 (H1N1)	1.6; 2011	[24]
DANA	2HTR	DANA—N8 complex; influenza A virus (unspecified strain)	2.5; 2006	[26]

**Table 2 viruses-15-01056-t002:** Synopsis of neuraminidase amino acids which interact with the five reference binders and the two hits. Residue numbering scheme adopted from PDB entry 3CL0. Residues are grouped to reflect their spatial packing and ordered from ionic to nonpolar. Abbr.: 3OHprop = tri-hydroxy-propyl; BB = peptide backbone; IA = interaction.

Residue	Sial ac.	DANA	Osel	Zana	Pera	Lani	AAmol	Fmol
Arg118	-COO (-)	-COO (-)	-COO (-)	-COO (-)	-COO (-)	-COO (-)	BB amide	BB amide
Arg292	-COO (-)	-COO (-)	-COO (-)	-COO (-)	-COO (-)	-COO (-)	-COO (-)	-COO (-)
Arg371	-COO (-)	-COO (-)	-COO (-)	-COO (-)	-COO (-)	-COO (-)	acetamido	BB amide
Arg152	acetamido	acetamido	acetamido	acetamido	acetamido	acetamido	-COO (-)	-COO (-)
Arg224	-	tri- hydroxy- propyl	-	-	-	-	-COO (-)	-COO (-)
Glu276	tri- hydroxy- propyl	tri- hydroxy- propyl	-	tri- hydroxy- propyl	-	-	-	-
Glu277	-	-	-	guanidino	Guanidino	guanidino	-	-
Glu119	-	-	-NH3 (+)	guanidino	Guanidino	guanidino	-	-
Asp151; -CH- not OO	2-hydroxy on oxane ring;	-	-NH3 (+); none	guanidino; none	guanidino; 2-hydroxy on cyclopentane	guanidino; none	amido; -S-CH3	piper- azinyl; none
Ser246	-	-	[no IA with alkyl ]	tri- hydroxy- propyl	-	2,3-dihydroxy- 1-methoxy propyl	-	-
Asn294	tri- hydroxy- propyl	same as left but Gly294 on N2 prot.	-	tri- hydroxy- propyl	-	2,3-dihydroxy −1-methoxy propyl	-	-
Tyr347			-COO (-)	-	-	-	acetamido	-
Tyr406	ether-O- in oxane	ether-O- in pyran	--	ether-O- in pyran	-	ether-O- in pyran	-	-
Val149 Ile 427 Pro431	-	-	-	-	-	-	phenyl	di- methyl- phenyl
Ala248	-	[no IA tri- hydroxy- propyl ]	-alkyl	-	2-ethylbutyl	-	-	-

**Table 3 viruses-15-01056-t003:** Estimated pharmacokinetic data for AAmol and Fmol. Abbreviations: BBB, blood–brain barrier; CYP, cytochrome P450 enzymes; MRTD, id, atom number of substrate; lethal doses; perm, permeability; sol, solubility % (m/m); Vd, volume of distribution; w, water. Asterisk (*****): even in its neutral form, Fmol remains hydrophilic, so its log P value becomes negative. Its intramolecular prototropy causes a zwitterionic form with total charges of +1 and −1 formally summating to zero. In contrast, AAmol can appear in a buffered solution as an undissociated (neutral) species with a positive log *p* value whichreflects its overall lipophilicity.

Name	Acidic pKa	MlogP (Neutral Form)	logD (ionized)	Perm Skin	Solu w
AAmol	3.8	1.0	−1.6	6.85	0.9
Fmol	11.3; 4.0	−1.6 (*)	−1.4 (*)	0.01	4.6
Name	pH in w	BBB_Filter	Vd in L/Kg	RuleOf5	CYP_1A2
AAmol	3.24	Low	0.22	0	No (96%)
Fmol	6.89	Low	0.54	0	No (96%)
Name	CYP_2C8	CYP_2C8 (id)	CYP_2C9	CYP_risk	TOX_MRTD
AAmol	Yes (73%)	S19(992); C20(869); C4(828)	No (56%)	0	Above_3.16
Fmol	No (92%)	NonSubstrate	No (98%)	0	Above_3.16
Name	TOX_hERG	TOX_ER	TOX_rat	TOX_skin	TOX_biodeg
AAmol	No (95%)	Nontoxic	2066.07	Nonsensit. (75%)	No (63%)
Fmol	No (95%)	Nontoxic	941.78	Nonsensit. (85%)	No (96%)

## Data Availability

Experimental results published at https://www.mdpi.com/1420-3049/25/18/4248 and at https://doi.org/10.3390/molecules25184248 (accessed on 17 April 2023). Patent document for AAmol at https://vidoc.impi.gob.mx/visor?usr=SIGA&texp=SI&tdoc=E&id=MX/a/2014/001768 (accessed on 20 April 2021). Patent document for Fmol at https://vidoc.impi.gob.mx/visor?usr=SIGA&texp=SI&tdoc=E&id=MX/a/2014/001759 (accessed on 20.04.2021).

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
