# Peer review of "Targeting the Human Influenza a Virus: The Methods, Limitations, and Pitfalls of Virtual Screening for Drug-like Candidates Including Scaffold Hopping and Compound Profiling"

_viruses, 2023, doi:10.3390/v15051056_

Round 1

Reviewer 1 Report

Scior et al. describes the in-silico screening and profiling of drug-like candidates. Using a 4D filtering method, the authors identified two potential lead structures as neuraminidase inhibitors starting from a virtual library containing >500.000 compounds. The study is well designed. However, the clarity of the presentation of the results is low and it is not so easy for the reader to follow their story. E.g., interactions were described in tables rather than illustrated in figures or the illustration of the binding mode of the known ligands has been exemplified for only one example or the quality of figures/structure formular could be improved. Moreover, the chemical IUPAC names should be carefully checked again for all compound, some contain typos or Spanish (?) language.

In my opinion, the manuscript is suitable for publication in Viruses after addressing the following comments:

1.) Abstract, line 15. The expression “some ligands” is too general. Please specify the number of known ligands.

2.) Page 2, lines 45/46: The expression “trail-and-error cycles” is in my opinion not true for any cases. Better use “optimization cycles”. By the way, virtual screenings and in silico profiling may also produce outcome data that do not correlate with the “wet” lab experiments.

3.) Page 2, line 62: Please correct “[[5 REFKCC 5]”

4.) Page 2, page line 97/98: The authors mention here known NA inhibitors. Please add the chemical structure of these NA inhibitors in a new figure. Also include the structure of sialic acid (Page 3, line 99).

5.) Page 3, lines 111-114. Please add the pdb codes for X-ray of the NA inhibitors here.

6.) Page 3, line 140: Ki should be defined by its first mentioning (Line 139) and the “i” should be subscript.

7.) Page 3, line 187: Please add bracket before “RB)”.

8.) Page 5, line 192: “Protonation” should start with a small letter.

9.) Page 5, line 236: sialic acid should be defined by its first mentioning (introduction)

10.) Pages 7 and 8, Table 2: In my opinion, this table is not so easy to interpret without presenting the binding mode of the ligands with the target NA. Please also add the corresponding figures in the SI part showing the interactions with the target (see also comment X,). Moreover, the abbreviation 3-OH-Prop is misleading (3-hydroxyproyl versus trihydroxypropyl) and this abbreviation is not used consequently in this table. Acetamido is written small and capital letter in the table.

11.) Page 10, Figure 3: Please also add 2D chemical structure for AAMol and FMol.

12.) Page 10, line 376: Please correct “dimetyl” to “dimethyl”

13.)  Page 11, line 428: What do the authors mean by “Ki = 1. uM” ?? and the “i” in Ki should be subscript.

14.) Page 11, ligand-target docking: In case of oseltamivir (line 410), an experimental Ki value is reported. The authors should compare their obtained in silico Ki values with the obtained or reported Ki values of the other known 4 ligands. Experimental Ki values of their two lead compounds AAMol and FMol should be also added to the manuscript.

15.) Page 11, line 434: CH2: the “2” should be subscript.

16.) Page 11, line 435: What to the author mean by carboxyl group (COO)? The carboxyl group (COOH) or the carboxylate (COO-)?

17.) Page 13, Table 3. The drug-likeness of the 2 lead structures has been predicted by in silico methods. Did the authors tested the two compounds in in vitro assays (CYP isoform inhibition assays, hERG assays etc.) to support the obtained data from the computational approach.

18.) SM, Binding pattern analysis. In Figure S1, the binding pattern of the known inhibitors is exemplified by only one compound. Please also add this analysis for the other ligands.

19.) SM, Figure S1. Please translate the Spanish language into English.

20.) SM, Table S3. Please use the English name for sialic acid.

21.) SM: Please also add binding pattern analysis for the AAMol and FMol.

22.) SM, ADMET profiling, output files: The quality of the chemical structures should be improved. It is not so easy to recognize all part/functional groups of the molecules.

Author Response

Please, see the attached file named 249_Comment2+reply_viruses-2237917.PDF. Thank you.

Reviewer 2 Report

The authors present methods for identifying drug-like molecules that are predicted to inhibit influenza viral neuraminidase from a large virtual library of small molecule compounds using computational screening, docking and profiling. The methods are well presented and could serve as a guide for rational virtual small molecule screening. However, there are several issues with the presentation of data and relevance of the findings that reduce enthusiasm for the manuscript (see below). Given that the biochemical effects of AAmol and Fmol small molecules are already published by the authors elsewhere (Reference 34, year 2020), this manuscript feels a bit of a hind-site. Perhaps the manuscript could be reworked into a ‘methods’ type manuscript for virtual screening.

Major Issues:

The introduction section is difficult to follow at times, in particular with regard to the function of viral NA as addressed below:

P2, line 54: unclear from the description what type of complexes are formed; do they include just NA and HA? Or are they part of a larger complex? What is the molar ratio of each moiety?

P2, line 55: ‘NA usually forms tetrameric complexes’ – is that separate from complex with HA?

P2, line 61: ‘only three circulate in the human body’ – which ones? Does that mean that influenza with only one of those three specific N subtypes can infect humans?

P2, line 73: ‘the last two sugar residues…’ – are the last two residues of what? It’s not clear from the description how the virus is attached to the cell.

P2, line 78: what is glycosidic bond ‘segregation’?

P2, line 89: Since both influenza strains (H1N1 and H5N1 subytpe) used in this study possess the N1 subtype, how is this virtual screening effort designed to identify molecules that can target a broad spectrum of influenza strains?

Table 1: Sialic Acid 3D structure (2BAT) is listed as being derived from a 3D SA-N1 3D complex, but is then identified as a H2N2 virus. Which is correct?

P7, line 260: A image that depicts the structural recognition of SA and inhibitors should be included in the manuscript instead of referencing figures in the literature.

P9, line 350: Likewise, predicted binding interactions of Fmol and AAmol with the N1 structure (2D and 3D images) should be shown with H-bonds, hydrophobic interactions and salt-bridges depicted between specific atoms of each small molecule and N1 sidechain/backbone atoms. The ‘final poses’ (referenced on p11, line 429) shown in Figure 4 and Figure 5 do not effectively illustrate the binding interactions between AAmol/Fmol and an N1 protein.

P9, line 356: The Ki and IC50 values (from the author’s reference #34) should be summarized here in a table. Text should describe specifically how small molecules with these binding parameters will advance the field of drug-discovery for influenza.

P10, line 369 and 373: redundant statements regarding AAmol structure.

P10, line 374: What is the take-away message from Reference 38 regarding AAmol as a peptide inhibitor?

P11, line 426-428: the authors state the affinities that AD4 (AutoDock version 4, I assume?) calculated for AAmol and Fmol. The values are much lower than the Ki values described in the author’s reference #34. A difference in experimental and calculated Ki is not unexpected but needs to be explained and described.

Minor Issues:

The manuscript is in need of editing for proper grammar usage and for concise, tighter language.

Author Response

Please, see attached file with my point-by-point response, named 249_Comment2+reply_viruses-2237917.pdf. Thank you.

Round 2

Reviewer 1 Report

The authors addressed all my comments during the revision process. In my opinion, the manuscript is now suitable for publication.

Author Response

Point-by-point response to the reviewer 2 = short cover letter
for
[Viruses] Manuscript ID: viruses-2237917 - Minor Revisions
ON BEHALF OF REVIEWER 2 :

PLEASE FIND OUR REPLY IN pdf FILE ATTACHED.

Reviewer 2 Report

The authors have addressed most of the concerns presented by this reviewer. However, several issues with the edited version of the manuscript. The manuscript lacks concise and clear language throughout and suffers ambiguity for some of the authors primary points because of that.   Further, the manuscript must be edited as despite what the authors claim as a more rigorous editing, the manuscript has many areas were proper grammatical usage of the English language is lacking.

1) The new title is not sound scientifically for a primary contribution.  The reviewer suggests a more engaging title that scientists will be draw to: ‘Methods, limitations and pitfalls of virtual screening for drug-like candidates including scaffold hopping and compound profiling: targeting the human influenza A virus’.  

2) The Introduction section will need to be modified to reflect a shift towards presenting methods/pitfalls for virtual screening, as the intent of the authors was indicated in their letter. On page 2, line 87, perhaps change the first sentence to: ‘The present study describes the methods for discovery of drug-like molecules using a combination of computational screening, docking and profiling. We also discuss some of the limitations of virtual screening and pitfalls to avoid during virtual screening. Our approach is exemplified by a description of our discovery of Fmol and AAmol small molecules that are predicted to inhibit budding of influenza viral particles.

3) P2, line 54 and line 55 (line #s from old version): The authors have not added clarity to the description of NA and HA proteins. The use of ‘or’ makes the statement about ‘NA or HA’ ambiguous. A better description of those proteins on the surface (how many of each, what they interact with, and how many variants of each is appropriate) must be written. If it is relevant to state that there are 500 NA (and it is still unclear if that means 500 tetramers or 125 tetramers) then it seems relevant to mention how many HA are present and whether they oligomerize. Similarly, M2 is stated as a surface protein but is not described at all. This leaves the reader unsure about its relevance. Does it need to be included? If so, what does M2 stand for? What is the function? Is it important for this manuscript?

4) P2, line 61 (line #s from old version): It is still unclear here. The authors state: ‘Of all nine existing NA subtypes, only influenza virus types A, B, and C circulate in the human body, in particular from influenza A virus our target protein N1’. This is not a sentence AND it is poorly constructed, conflating two separate ideas: viral subtypes and specific protein subtypes. Is N1 only expressed by influenza A? Which HA does influenza A express? Disambiguation of viral and protein subtypes is needed and the connection between the two (if any) needs to be better explained. This is also a good example of why the manuscript would be aided by editing for proper use of the English language.

5) P2, line 77-78 (line #s from new version): this is still confusing: What part of the virus is attached to the cell glycoprotein? Which glycoprotein? A better description is needed here.

6) P10, line 372: there is no need to mention it is ‘three digit’ because the authors state the Ki specifically in the same sentence.

7) Figure 4 and 5: Remove ‘Graphical analysis’ and replace with ‘Structural models’. Change ‘binding modes’ to  ‘predicted binding modes’.

Author Response

Please, see the attached PDF file.
